# D5RL: DIVERSE DATASETS FOR DATA-DRIVEN DEEP REINFORCEMENT LEARNING

## ABSTRACT

Offline reinforcement learning algorithms hold the promise of enabling data-driven RL methods that do not require costly or dangerous real-world exploration and benefit from large pre-collected datasets. This in turn can facilitate real-world applications, as well as a more standardized approach to RL research. Furthermore, offline RL methods can provide effective initializations for online finetuning to overcome challenges with exploration. However, evaluating progress on offline RL algorithms requires effective and challenging benchmarks that capture properties of real-world tasks, provide a range of task difficulties, and cover a range of challenges both in terms of the parameters of the domain (e.g., length of the horizon, sparsity of rewards) and the parameters of the data (e.g., narrow demonstration data or broad exploratory data). While considerable progress in offline RL in recent years has been enabled by simpler benchmark tasks, the most widely used datasets are increasingly saturating in performance and may fail to reflect properties of realistic tasks. We propose a new benchmark for offline RL that focuses on realistic simulations of robotic manipulation and locomotion environments, based on models of real-world robotic systems, and comprising a variety of data sources, including scripted data, play-style data collected by human teleoperators, and other data sources. Our proposed benchmark covers state-based and image-based domains, and supports both offline RL and online fine-tuning evaluation, with some of the tasks specifically designed to require both pre-training and fine-tuning. We hope that our proposed benchmark will facilitate further progress on both offline RL and fine-tuning algorithms. Website with code, examples, tasks, and data is available at https://sites.google.com/view/d5rl/

## 1    INTRODUCTION

Offline reinforcement learning algorithms hold the promise of enabling data-driven RL methods that do not require costly or dangerous real-world exploration, and benefit from pre-collected datasets (Levine et al., 2020; Gulcehre et al., 2020; Agarwal et al., 2020). The latter especially is of significant relevance in the modern age of data-driven machine learning, where training on large datasets has repeatedly been shown to be a critical ingredient for effective generalization (LeCun et al., 2015; Krizhevsky et al., 2017) and even emergent capabilities (Wei et al., 2022). Furthermore, offline RL methods can provide effective initializations for online finetuning, overcoming challenges with exploration and providing an effective formula for fast online training suitable for the real world. However, while supervised learning methods that operate on large pre-collected datasets can effectively evaluate on test sets sampled from real-world data, offline RL algorithms that train on data must still be *validated* through online interaction to measure their effectiveness, even if no online interaction is required during training. Therefore, evaluating progress on offline RL methods requires effective and challenging benchmarks that can provide for accessible evaluation in simulation, while still providing a degree of realism in terms of reflecting the properties of real-world systems, and covering a range of challenges both in terms of the parameters of the domain (e.g., length of the horizon, sparsity of rewards) and the parameters of the data (e.g., narrow demonstration data or broad exploratory data). Existing benchmarks have enabled significant advances in offline RL in recent years. However, these are largely simple environments, which might fail to reflect properties of realistic tasks, and might not cover some of the most significant use cases, such as online finetuning from offline initialization  (Fu et al., 2020; Gulcehre et al., 2020). Moreover, new algorithms are

increasingly saturating in performance, indicating that we might be approaching the limits of these datasets.

In this paper, we propose a new benchmark for offline RL that focuses on realistic simulations of robotic manipulation and locomotion environments, based on models of real-world robotic systems, and comprising a variety of data sources, including scripted data, play-style data collected by human teleoperators, and other data sources. Our proposed benchmark covers state-based and image-based domains, and supports both offline RL evaluation and evaluation with online finetuning, with some of the tasks specifically designed to require both pretraining and finetuning. We hope that our proposed benchmark will facilitate further progress on both offline RL algorithms and algorithms designed for online finetuning from offline initialization.

We present an overview of the environments in our benchmark, which include realistic simulated models of real-world robotic platforms, such as the A1 quadruped and the Franka robotic arm. Aside from providing a more challenging and up-to-date range of tasks and datasets compared to prior work (Fu et al., 2020; Gulcehre et al., 2020), our tasks cover a range of factors that are either rarely covered in prior benchmarks, or rarely appear in combination. The A1 tasks specifically evaluate online finetuning: these tasks are designed such that offline initialization should provide for basic but low-performance capability (e.g., not falling), while online finetuning is required for maximally effective gaits. The visual Franka kitchen environments evaluate visual perception, environment variability (accomplished via randomization), and ability to use "play-style" diverse data collected by real humans via teleoperation. The visual WidowX pick-and-place environments evaluate the ability to "stitch together" distinct phases of manipulation skills to accomplish multi-stage behaviors. While prior datasets evaluate stitching (e.g., the AntMaze task in D4RL (Fu et al., 2020)), it is rarely evaluated in combination with visual perception in widely accepted benchmarks.

We provide a comprehensive description of our proposed tasks and corresponding datasets, as well as high-quality implementations of a number of widely used offline RL and online finetuning methods that we evaluate on our benchmark. We hope that this will provide a solid foundation for future progress on both offline reinforcement learning and online finetuning from offline initialization.

## 2 RELATED WORK

Benchmarking in reinforcement learning has been a persistent challenge, with effective benchmarks needing to balance accessibility (i.e., tasks that are feasible to address with current methods and not too onerous computationally) with the desire for broad coverage of task properties and a high degree of realism and complexity (Duan et al., 2016; Brockman et al., 2016; Wu et al., 2017; Wang et al., 2019; Hubbs et al., 2020; Yu et al., 2020). Striking this balance is arguably a greater challenge in RL than in other fields. First, RL algorithms can be applied to a wide range of tasks with very different properties, including varying time horizons, levels of reward sparsity, dimensionality, and other ingredients (Osband et al., 2019). Second, RL algorithms can be computationally very demanding, requiring long training runs that make it difficult to include large numbers of very complex tasks in every evaluation (Henderson et al., 2018; Agarwal et al., 2021). Third, the capabilities of RL methods have advanced significantly over the past decade, and benchmarks can quickly become saturated, necessitating more complex tasks to be added (Dulac-Arnold et al., 2021). This makes designing a good benchmark in RL a major challenge. Our work focuses specifically on benchmarking offline RL methods, and aims to strike a balance between covering task complexity and a variety of task ingredients with providing a convenient simulated evaluation protocol and a mixture of image-based and state-based tasks.

In recent years, a number of benchmarks have been proposed for offline RL, though such benchmarks typically have a number of shortcomings that have proven difficult to fully alleviate while balancing the aforementioned challenges. Early work on deep offline RL focused either on customized evaluations without proposing standard benchmarks (Vecerik et al., 2017; Hester et al., 2018; Kalashnikov et al., 2018), or else proposed simple benchmark tasks that utilized replay buffers from successful RL runs (Fujimoto et al., 2019; Kumar et al., 2019; Agarwal et al., 2020). The latter generally does not evaluate the performance of offline RL methods effectively, as realistic data might be highly suboptimal and might require "stitching" together parts of different suboptimal trajectories to create ones that are more optimal – a property rarely captured by data collected by fully or partially trained RL policies themselves (Fu et al., 2020; Levine et al., 2020). Several more recent offline RL benchmarks have sought to include more realistic data distributions, more complex tasks (including

vision-based tasks), and other ingredients that are intended to more accurately represent realistic offline RL problems (Gulcehre et al., 2020; Liu et al., 2022; Kurenkov and Kolesnikov, 2022; Kuo et al., 2022; Qin et al., 2022; Lu et al., 2022). Some works have proposed protocols for benchmarking offline pretraining with online finetuning (Kostrikov et al., 2021; Nair et al., 2020; Song et al., 2022; Nakamoto et al., 2023), though this has not been rigorously systematized in prior work. Perhaps the most widely used benchmark suite in offline RL today is D4RL (Fu et al., 2020). However, the D4RL tasks are increasingly saturated in performance, and many of the tasks do not effectively reflect the challenges of realistic offline RL tasks: the MuJoCo locomotion tasks in D4RL are still largely based on RL replay buffers, and the more complex "maze" tasks, which do feature suboptimal data and require stitching or recombining parts of the suboptimal trajectories, are limited in difficulty and variety. Our benchmark aims to address these limitations in several ways. We focus specifically on robotics-themed tasks – although RL can address a far greater range of problems, we believe that this focus is reasonable for providing a balance between specificity (i.e., not so much breadth that no single method can address all tasks) and coverage (i.e., still capturing different challenges in RL). Within this theme, our tasks all reflect realistic simulated models of robotic systems based on actual robot URDF specifications, in contrast to D4RL, which uses simple "fictional" rigid body systems. Our tasks include both state-based and image-based tasks, both sparse and dense rewards, and multi-stage tasks. Additionally, we propose tasks suitable for offline pretraining with online finetuning, something that has not been rigorously formalized in current widely used benchmarks.

Offline RL algorithms themselves have made significant progress in recent years as well (Fujimoto et al., 2019; Kumar et al., 2019; 2020; Agarwal et al., 2020; Kostrikov et al., 2021; Nair et al., 2020; Song et al., 2022; Cheng et al., 2022; Nakamoto et al., 2023). A full survey of all recent research on offline RL is outside the scope of this paper, but we do make an attempt to benchmark representative examples of some of the widely used algorithm classes, including pessimistic or conservative algorithms (Kumar et al., 2020; Nakamoto et al., 2023), algorithms based on implicit backups (Kostrikov et al., 2021), algorithms based on behavioral cloning regularization (Fujimoto and Gu, 2021), algorithms that utilize diffusion models (Hansen-Estruch et al., 2023), and methods designed specifically for efficient online training by leveraging offline data (Ball et al., 2023). We hope that by proposing a new benchmark that addresses the limitations of prior datasets and environments we will provide a more effective means for algorithms researchers to make further advances in the future.

## 3 PRELIMINARIES AND BACKGROUND

Reinforcement learning is formalized through the concept of Markov Decision Process (MDP) $\mathcal{M} = (\mathcal{S}, \mathcal{A}, P, R, \rho, \gamma)$, where $\mathcal{S}$ is the state space, $\mathcal{A}$ is the action space, $P(s'|s,a)$ is the transition probability, $R(s,a)$ is the reward function, $\rho$ is the initial state distribution and $\gamma$ is a discount factor. The goal of reinforcement learning is to find a policy $\pi(a|s)$ that maximizes the expected reward:

$$J(\pi) = \mathbb{E}_{\rho, P, \pi} \left[ \sum_{t=0}^{\infty} \gamma^t R(s_t, a_t) \right] \tag{1}$$

In the standard RL setting the policy is given access to the MDP and can sample trajectories to collect additional data. On-policy algorithms iterate between data collection and policy updates, and discard the collected data after each update, which makes them sample inefficient. Off-policy algorithms collect data in a replay buffer, which is then repeatedly used to update the policy.

*Offline reinforcement learning* also reuses previously collected data, but unlike off-policy algorithms it does not have access to the MDP during training and only utilizes a static dataset. These algorithms need to be able to handle distribution shift between their training datasets and deployment. Moreover they need to be able to utilize a variety of data sources and qualities, such as prior training runs, deployments, data from different agents or human-generated data.

Additionally, prior offline data can be leveraged with online RL, either by *pretraining* offline and *finetuning* online (Nair et al., 2020; Kostrikov et al., 2021), or by training online but including the prior data in a replay buffer (i.e., joint offline and online training) (Song et al., 2022; Ball et al., 2023). The challenge in this setting is for the policy to effectively utilize the offline data to reach high performance in a sample-efficient way.

Our proposed tasks and datasets can be used for both problems, pure offline RL and offline-to-online fine-tuning, and we evaluate both settings in our experiments.

## 4    CHALLENGES IN OFFLINE RL EVALUATION

Our benchmark environments and datasets aim to cover a range of challenges that are likely to be encountered by offline RL algorithms aiming to learn effective policies for real-world tasks. Some of these challenges, like temporal compositionality ("stitching"), have been addressed via simpler and less realistic environments in prior benchmarks (Fu et al., 2020). Other challenges, like the use of visual observations, are present in prior tasks (Gulcehre et al., 2020), but in combination with less realistic data distributions, such as data from the replay buffer of online RL runs. We discuss some of these challenges below, and in Section 5 discuss how our tasks instantiated some of these challenges.

**Datasets:** The performance of offline RL methods is strongly dependent on the data distribution. Therefore, to provide a comprehensive benchmark for offline RL, we include a variety of different challenging yet realistic distributions in our tasks. This includes narrow distributions from scripted planners and human-generated demonstration data. While some prior benchmarks also include scripted and human-generated data (Fu et al., 2020), many of the previously studied tasks consist of replay buffers from online RL runs (Gulcehre et al., 2020), which may not be reflective of the data distributions on which we might want to train real-world systems. In our WidowX-based long-horizon tasks, we generate object manipulation data using (sub-optimal) scripted planners. In the Franka domain, we collected 20 hours of new human teleoperation data, and also include tasks based on human teleoperation datasets from prior work (Gupta et al., 2019a; Fu et al., 2020), but rendered out with visual observations rather than low-dimensional state. We include both expert-level demonstrations from an experienced teleoperator, as well as play data from several teleoperators with different levels of experience. We believe that these data distributions are realistic, in the sense that they reflect data sources that might actually be used for real-world training, and challenging for current methods.

**Temporal compositionality and multi-stage tasks:** One of the most appealing properties of offline RL methods is the ability to combine parts of suboptimal behaviors and compose them into new behaviors that complete more complex tasks more effectively (Levine et al., 2020; Fu et al., 2020). One of the ways that offline RL can do this is by exploiting temporal compositionality: if the algorithm understands that it's possible to reach C from B, and to reach B from A, then it should be able to figure out how to reach C from A. This can enable solving multi-stage tasks (such as sorting multiple objects) by composing shorter-horizon primitive behaviors. Our benchmarks are designed to evaluate temporal compositionality both by composing task-agnostic or multi-task suboptimal data (e.g., "play" data) into longer and more optimal tasks, and by composing single-step behaviors to solve multi-stage tasks, such as sorting objects.

**Online training from offline data:** In many cases, we might want to use offline RL not to acquire a policy that we deploy in the real world in zero shot, but rather to provide an initialization for online training for a skill that would be difficult (or dangerous) to acquire entirely from scratch. This can be done either via offline pretraining and finetuning Nair et al. (2020); Kostrikov et al. (2021); Nakamoto et al. (2023), or by using online RL algorithms that can incorporate offline data (Song et al., 2022; Ball et al., 2023). Prior benchmarks rarely evaluate this setting, and prior works studying this setting tend to use a non-standard combination of tasks adopted by the community.

**Realistic observation spaces:** Previous offline RL benchmarks, such as D4RL Fu et al. (2020), mostly focus on low-dimensional state observations. However, scaleable RL algorithms should be able to learn from high-dimensional observations, such as RGB images. We therefore include image observations in many of our tasks, reflecting realistic observation spaces for each domain.

**Diverse and realistic robot systems:** Our benchmark environments are themed around a variety of robotic learning domains. While such simulated continuous control tasks are common in prior benchmarks (Brockman et al., 2016; Gulcehre et al., 2020; Fu et al., 2020), we directly use realistic simulations of real-world robotic systems, including the A1 quadruped and WidowX and Franka Emika robotic arms. All of the robots are based on their actual URDFs (definitions of robot morphology), controlled in ways that are analogous to their real-world counterparts (e.g., position control or end-effector control).

**Generalization to initial conditions:** One of the central challenges for real-world RL systems is to handle generalization to task and domain variability. However, prior benchmarks in RL often do not emphasize generalization to different initial conditions. To evaluate agent's robustness and ability to generalize, some of our tasks vary the objects the robot needs to manipulate and randomize their arrangement. In addition, on the observation side we introduce a number of distractors by varying textures, object colors, lighting conditions and camera angles.

# 5 D5RL: DIVERSE DATASETS FOR DATA-DRIVEN DEEP REINFORCEMENT LEARNING

In this section, we describe the individual tasks in our benchmark, and relate them to the challenges outlined in the preceding section. Each of our tasks reflects a realistic simulated model of a robotic system, using the URDF of the corresponding robot and a simulated environment to enable plausible interactions. Although our goal is primarily to enable rapid algorithms development rather than to provide a framework for robotics research specifically, we believe that this added degree of realism increases the chances that algorithmic developments made with our benchmark will translate into good real-world performance, and that the challenges presented in our tasks also reflect other domains beyond robotics. This includes narrow data distributions (common for systems where data is collected from baseline hand-designed controllers or humans), high-dimensional observations, the need for online finetuning, and temporal compositionality. Beyond the below descriptions, additional details about the environments and datasets are provided in Appendix A.

## 5.1 LEGGED LOCOMOTION

The goal of the legged locomotion tasks is to study the efficacy of offline RL methods in handling low-level control problems with complex dynamics. We set up these tasks on a simulated Unitree A1 robot platform and require learning policies from low-dimensional proprioceptive observations and do not require visual perception.

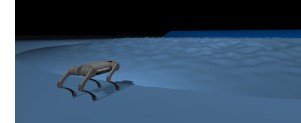

Figure 1: **Hiking task**. The A1 robot at the start of the course in front of a randomized terrain.

**Tasks:** Concretely, we construct three offline datasets, each of which aim to learn different types of locomotion skills as follows:

1. **Interpolate Speed:** The goal is to control the A1 at a particular speed level, within the range of speeds that were observed in the training data. For this, we first collect a dataset by training an A1 to track 3 speeds: 0.5, 0.8, and 1.0 m/s, containing experience from the agents' initial exploration to expert-level performance on those tasks, and the goal is to adapt to a speed value of 0.75 m/s, that lies within the range of speeds observed in the dataset. To compute rewards for offline RL training, we label each transition with how accurately it tracks the target speed of 0.75 m/s.

2. **Extrapolate Speed:** Using the same dataset as the Interpolate Speed task, this task instead tests the ability of an algorithm to be able to acquire a policy that can run at a higher speed of 1.25 m/s. This task presents a challenge for offline RL methods as the optimal policy that runs at the higher speed lies outside the support of the offline dataset, which means that this task presents a significant room for improvement with online fine-tuning.

3. **Hiking:** Finally, we construct a task that aims to test the efficacy of offline RL at learning policies when interacting with the complex dynamics induced by navigating on a hiking course (shown in Fig. 5.1). This task still utilizes a offline dataset that depicts navigation on a flat terrain, but is distinct in that the policy is deployed on a hiking course, and not a flat terrain. Our hiking course presents varied terrains consisting of randomly generated rolling bumps as well as inclines, and the goal is to navigate the policy to the center of the course without falling.

## 5.2 FRANKA KITCHEN MANIPULATION ENVIRONMENT

The goal of this environment is to study offline RL and online fine-tuning from realistic but suboptimal human-generated data, evaluate settings with variability in the appearance and placement of objects to measure generalization, and handle multiple visual observations. Near-optimal and sub-optimal human-collected data, which can run the gamut from demonstrations to unstructured "play", represents a realistic source of training data for offline RL, which has been studied in several prior works (Lynch et al., 2020; Gupta et al., 2019a; Mandlekar et al., 2021). Additionally, generalization over object placement and appearance is very important in real-world settings, but is rarely evaluated in RL benchmarks (Cobbe et al., 2019; 2020). Therefore, we hope that this task will cover a range of

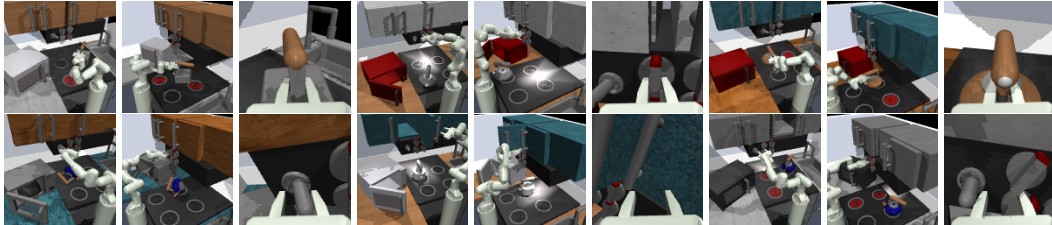

Figure 3: Observations from the Randomized Kitchen environment consist of two $128 \times 128$ RGB images from side-cameras, $128 \times 128$ RGB image from a wrist camera, and robot proprioception. The environment includes several different types of kettles and microwaves, which require different grasps. Moreover, their locations are randomized across the scene. Textures, lighting conditions, and camera angles are also varied across episodes.

challenges that are underrepresented in prior work. This environment consists of a Franka Emika robot in a simulated kitchen setting, and data is collected via VR-based tele-operation by real people. We introduce several environments that pose different challenges for current data-centric RL algorithms.

### 5.2.1 STANDARD FRANKA KITCHEN ENVIRONMENT

For an easier starting point, we adapt the Franka Kitchen environment which was introduced by Gupta et al. (2019a) and was also part of the D4RL (Fu et al., 2020) benchmark. The objective in this environment is to manipulate a set of 4 pre-specified objects. We modify the task to utilize multiple image observations rather than ground truth object locations, thus providing an observation space that more realistically reflects robotic manipulation scenarios. The agent receives a sparse reward of +1 for every object manipulated into the correct configuration.

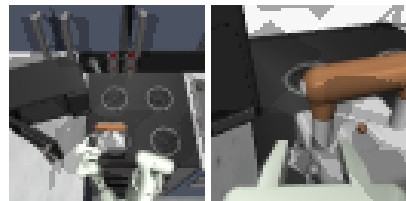

Figure 2: Observations for the Standard Franka Kitchen tasks consist of two $64 \times 64$ RGB images from an a top-down and a wrist camera, as well as robot proprioception.

**Datasets:** We use the same datasets as (Gupta et al., 2019b; Fu et al., 2020), which consists of expert-level demonstrations for different combinations of four objects, executed in a fixed order. In total there are 513 total trajectories of varying length split across 22 task combinations. Our observation space consists of two $64 \times 64$ images from a side-view and wrist cameras Hsu et al. (2022) as shown in Fig. 2, as well as robot proprioception.

**Tasks**: We consider two settings, similar to Fu et al. (2020):

1. **Mixed:** In this environment the agent needs to rearrange the microwave, kettle, light switch and slide door objects, and there are several expert demonstrations in the offline dataset for that combination of objects.

2. **Partial:** In this setting the agent needs to manipulate the microwave, kettle, bottom burner knob and light switch objects, which are never encountered together in any of the trajectories in the offline dataset. This requires the agent to learn combinatorial generalization capabilities. We note that this is different from the dynamic programing or "stitching" probalem, since there is no sequence of states in the dataset that reach the optimal solution.

### 5.2.2 RANDOMIZED FRANKA KITCHEN ENVIRONMENT

We include a version of the Franka Kitchen environment with randomized scene configurations to further test generalization. The environment was constructed by modifying the "Kitchenshift" domain Xing et al. (2021). Both object types and their locations in the environment are randomized, which requires the agent to learn robust and general grasping strategies. There are several types of visual distractors, including randomized textures and lighting conditions. The observation space consists of three $128 \times 128$ images: two side-view cameras and a wrist camera, as well as robot proprioception. The exact camera positions are also continuously randomized. Observations from different episodes are included in Fig. 3. This level of variability introduces a significant challenge in terms of robustness and representation learning, reflecting challenges likely to be seen in the real world.

**Datasets**: To provide offline training data in this domain, we manually collected close to 20 hours of human teleoperation data:

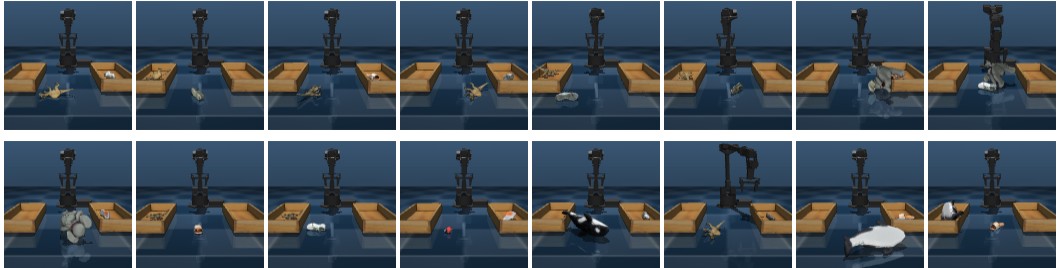

Figure 4: Observations from the Multi-Stage Manipulation environments consist of a single $128 \times 128$ RGB image from a side camera and robot proprioception. The environment includes several different types of shoes and toys, which require different grasps. Moreover, their locations are randomized across the scene. Textures, lighting conditions, and camera angles are also varied across episodes.

1. **Demonstrations:** We collected 500 expert-level demonstrations from an experienced teleoperator for the microwave, kettle, light switch and slide cabinet task (the same as the "Mixed" dataset from Section 5.2.1). This dataset is suitable for testing capabilities of representation learning approaches and benchmarking imitation learning algorithms.

2. **Play:** We collect a datasets of 1000 episodes, which are not task-oriented from multiple operators with different levels of skills. The episodes consist of undirected environment interactions and involve manipulating between 2 to 6 objects in random order and placement. These episodes were collected by several tele-operators with different levels of experience, which introduces significant multi-modality in the data both in terms of behaviours and quality of executed grasps.

3. **Sub-optimal Expert:** We also include a sub-optimal expert dataset consisting of 500 episodes, collected by an inexperienced teleoperator, but we do not explicitly benchmark it in this work.

**Tasks:** On the Demonstrations dataset, the agent is evaluated on the task corresponding to that demonstration. On the Play dataset, similar to Section 5.2.1, we consider two tasks:

1. **Mixed:** Similar to before in this task we need to manipulate the the microwave, kettle, light switch and slide cabinet objects. However, in addition to the representation learning and robustness challenges that the randomized kitchen poses, the agent needs to learn from diverse data of varying quality. Another challenge is that while there are several episodes which manipulate all four objects, they do so in a different order, which creates a challenging problem for dynamic programming with multi-modal solutions.

2. **Partial:** Similar to before, the agent needs to manipulate the microwave, kettle, bottom burner knob and light switch objects, which are never solved in the same episode in the offline data.

### 5.3 Multi-Stage Manipulation with Scripted Data

The goal of this task is to study composition of suboptimal trajectories to solve longer-horizon tasks, incorporate visual observations, and handle data from weak scripted policies. These ingredients reflect problems that are often encountered in offline robotic RL, where we might want to compose longer-horizon behaviors out of datasets depicting individual primitive skills (Fang et al., 2022; Rosete-Beas et al., 2023; Fang et al., 2023). To this end, we introduce a multi-stage bin sorting task. The simulated robot is a 6-DOF WidowX arm placed in front of two identical white bins with 2 objects to sort. These two objects are from two different categories: shoes and toys, and are taken from the Google Scanned Objects Dataset (Downs et al., 2022), comprising 3D scans of real household objects.

**Task:** The environment is shown in Figure 4. The objective is to sort each object into its respective bins. One bin corresponds to shoes and the other bin corresponds to toys. The reward function is the number of objects correctly sorted into each bin, where a "+1" reward is given when any of the objects are placed in their correct bins and a "+2" reward is given when both objects are sorted correctly. The observations consist of the proprioceptive state of the robot (joint positions) and $128 \times 128 \times 3$ image observations. The initial state is randomized in terms of the objects. At each environment reset, one toy and one shoe are randomly selected from a pool of 5 objects and placed in the central region of the scene. Various sample image observations can be seen in Figure 4.

| Environment | Task | Method | | | | | | |
|---|---|---|---|---|---|---|---|---|
| | | BC | IQL | CQL | CalQL (Nakamoto et al., 2023) | TD3 + BC | DDPM + BC | IDQL |
| Standard Kitchen | Mixed | $0.461 \pm 0.124$ | $0.457 \pm 0.129$ | $0.0 \pm 0.0$ | $0.0 \pm 0.0$ | $0.003 \pm 0.003$ | $0.253 \pm 0.082$ | $0.020 \pm 0.008$ |
| | Partial | $0.474 \pm 0.063$ | $0.427 \pm 0.116$ | $0.0 \pm 0.0$ | $0.0 \pm 0.0$ | $0.053 \pm 0.075$ | $0.163 \pm 0.054$ | $0.087 \pm 0.021$ |
| Randomized Kitchen | Demos | $0.144 \pm 0.010$ | $0.174 \pm 0.031$ | $0.023 \pm 0.032$ | $0.023 \pm 0.016$ | $0.052 \pm 0.033$ | $0.126 \pm 0.016$ | $0.033 \pm 0.011$ |
| | Mixed | $0.057 \pm 0.019$ | $0.027 \pm 0.0$ | $0.005 \pm 0.002$ | $0.004 \pm 0.001$ | $0.057 \pm 0.026$ | $0.105 \pm 0.016$ | $0.009 \pm 0.004$ |
| | Partial | $0.072 \pm 0.019$ | $0.048 \pm 0.015$ | $0.003 \pm 0.005$ | $0.001 \pm 0.001$ | $0.023 \pm 0.007$ | $0.044 \pm 0.010$ | $0.002 \pm 0.001$ |
| Locomotion | a1-walk-v0 | $1.006 \pm 0.015$ | $0.962 \pm 0.007$ | $0.068 \pm 0.112$ | $-0.171 \pm 0.033$ | $0.549 \pm 0.178$ | – | – |
| | a1-run-v0 | $0.684 \pm 0.026$ | $0.932 \pm 0.006$ | $-0.067 \pm 0.045$ | $-0.206 \pm 0.086$ | $0.002 \pm 0.021$ | – | – |
| | a1-hiking-v0 | $0.956 \pm 0.004$ | $0.935 \pm 0.003$ | $0.0 \pm 0.004$ | $-0.013 \pm 0.008$ | $0.003 \pm 0.001$ | – | – |
| WidowX | wx-sorting-v0 | $0.152 \pm 0.032$ | $0.021 \pm 0.016$ | $0.0 \pm 0.0$ | $0.0 \pm 0.0$ | $0.016 \pm 0.022$ | $0.041$ | $0.173$ |
| | wx-sorting-pickplacedata-v0 | $0.084 \pm 0.048$ | $0.0 \pm 0.0$ | $0.0 \pm 0.0$ | $0.0 \pm 0.0$ | $0.0 \pm 0.0$ | $0.081$ | $0.25$ |

Table 1: **Offline evaluation** of each task, dataset, and method. The numbers shown are the normalized returns evaluated at the end of 500k gradient steps of offline training averaged over three random seeds. For the Standard Kitchen and the Randomized Kitchen environments, the return corresponds to how many of the four target objects are successfully manipulated over the course of an episode (ie: a score of 1.0 indicates that all four objects were successfully manipulated). For the Locomotion and WidowX environments, the scores are normalized between a random policy and an expert policy (ie: a score of 0 corresponds to obtaining the same return as a fully random policy, and a score of 1 corresponds to obtaining the same return as the expert policy). The expert policy is obtained by training an SAC agent online to convergence.

| Environment | Task | Method | | | | | | |
|---|---|---|---|---|---|---|---|---|
| | | IQL | CQL | CalQL (Nakamoto et al., 2023) | TD3 + BC | RLPD (Ball et al., 2023) | DDPM + BC | IDQL |
| Standard Kitchen | Mixed | $0.123 \pm 0.102$ | $0.0 \pm 0.0$ | $0.0 \pm 0.0$ | $0.067 \pm 0.066$ | $0.139 \pm 0.075$ | $0.200 \pm 0.029$ | $0.020 \pm 0.008$ |
| | Partial | $0.290 \pm 0.064$ | $0.0 \pm 0.0$ | $0.0 \pm 0.0$ | $0.093 \pm 0.059$ | $0.221 \pm 0.113$ | $0.177 \pm 0.022$ | $0.087 \pm 0.021$ |
| Randomized Kitchen | Demos | $0.234 \pm 0.017$ | $0.0 \pm 0.0$ | $0.023 \pm 0.016$ | $0.052 \pm 0.033$ | $0.001 \pm 0.001$ | $0.166 \pm 0.029$ | $0.033 \pm 0.011$ |
| | Mixed | $0.25 \pm 0.0$ | $0.0 \pm 0.0$ | $0.004 \pm 0.001$ | $0.057 \pm 0.026$ | $0.011 \pm 0.009$ | $0.133 \pm 0.004$ | $0.009 \pm 0.004$ |
| | Partial | $0.021 \pm 0.009$ | $0.0 \pm 0.0$ | $0.001 \pm 0.001$ | $0.023 \pm 0.007$ | $0.0 \pm 0.0$ | $0.084 \pm 0.009$ | $0.002 \pm 0.001$ |
| Locomotion | a1-walk-v0 | $0.935 \pm 0.017$ | $0.068 \pm 0.112$ | $0.750 \pm 0.027$ | $0.030 \pm 0.003$ | $1.016 \pm 0.005$ | – | – |
| | a1-run-v0 | $0.936 \pm 0.021$ | $-0.067 \pm 0.045$ | $0.700 \pm 0.066$ | $0.110 \pm 0.091$ | $1.011 \pm 0.007$ | – | – |
| | a1-hiking-v0 | $0.927 \pm 0.014$ | $0.0 \pm 0.004$ | $0.368 \pm 0.107$ | $0.938 \pm 0.015$ | $1.058 \pm 0.020$ | – | – |

Table 2: **Offline-to-online evaluation** of each task, dataset, and method. The numbers shown are the normalized returns evaluated at the end of 500k additional gradient steps of online finetuning averaged over three random seeds. See Table 1 for how normalized scores are computed.

**Datasets:** There are 2 tasks corresponding to the environments wx-sorting-v0 and wx-sorting-pickplacedata-v0. Below, we provide a description of each.

1. **Sorting:** The first dataset comprises of data collected with a scripted policy that attempts sorting both objects into their respective bins. The scripted policy with some likelihood places the object in the correct bin if grasped and otherwise in the incorrect bin. In all, there are 2000 episodes presented to the agent, which are mostly unsuccessful at solving the full task but consistently solve the individual segments of the task in separate episodes.

2. **Sorting with Pickplace Data:** This dataset only comprises of transitions where the robot picks any object and places it in its respective bin. The data is similar to the dataset above in that there is a likelihood that the scripted agent places the object in the wrong bin. In all, there are 2000 episodes presented to the agent, solving the sorting task only partially.

## 6 BENCHMARK RESULTS

For each of the datasets in each of the domains, we evaluated a collection of recently proposed offline RL algorithms, as well as methods designed for online RL training with offline data (either via pretraining or joint training). We selected a range of algorithms that are meant to be representative of various different types of approaches. Although our evaluation algorithms do not cover every recent method (as there are many of them), we evaluated 8 separate algorithms, and we hope that in collaboration with the community, we can include many more evaluation numbers as part of the D5RL open-source repository. We chose CQL (Kumar et al., 2020) as a standard representative example of a pessimistic/conservative offline RL method, together with Cal-QL (Nakamoto et al., 2023), a variant of CQL adapted for online finetuning. To evaluate implicit TD backups, we include IQL (Kostrikov et al., 2021), as well as IDQL (Hansen-Estruch et al., 2023), a recent extension of IQL that utilizes diffusion model policies. To evaluate BC-based regularization, we include TD3+BC (Fujimoto and Gu, 2021). We include RLPD (Ball et al., 2023) as a representative example of a joint training method

that runs online RL with prior data included in the buffer, and a behavioral cloning (BC) baseline as a diagnostic of the average performance in the dataset.

The results for all of the offline RL methods are included in Table 1, with results after online finetuning included in Table 2. For completeness, we include RLPD in the offline results (using the same exact algorithm but without online collection). The online results are obtained by finetuning the offline value function and policy for each method, except for RLPD, where the online run is completely separate from the offline one. Further details about the specific training setup, hyperparameters, and number of update steps for each method are provided in Appendix B.

The results show that our proposed benchmark leaves considerable room for improvement for current offline RL and online finetuning methods. A few particularly prominent challenges include handling generalization and visual observations, and handling multi-stage tasks. When using image observations for the Franka kitchen tasks, particularly the more complex randomized domain, we see that many of the current RL methods struggle to exceed the performance of the simple behavioral cloning policy, indicating significant difficulties in learning robust perception. When learning the multi-stage WidowX tasks, we similarly see low performance, and in fact the naïve BC policy performs marginally better, again suggesting difficulties with scaling current RL methods into these domains. We believe that these results indicate that our benchmark provides significant room for improvement, and can drive development of more effective and scalable methods.

## 7 DISCUSSION

We introduced a new benchmark for offline RL and online training with offline data, which we call D5RL. The aim of D5RL is to provide coverage of a variety of offline RL and online finetuning challenges, including different data compositions (scripted, human play-style data, and other sources), different input modalities (images and state), and tasks that require varying degrees of stitching, online finetuning, and generalization over initial state variability. Although the D5RL tasks are designed primarily for iterating on RL algorithms, all of the D5RL tasks are also designed to be reasonably reflective of real-world robotic tasks, with each environment containing a simulation of a real-world robot (an A1 quadruped, a Franka industrial arm, or a WidowX low-cost robotic arm) based on the robot's actual URDF, and tasks that reflect behaviors those robots might be expected to carry out in the real world. We also conducted an investigation with a number of existing offline RL and online finetuning methods to provide initial evaluation numbers with our benchmark, which we hope the community will utilize to develop more effective algorithms.

While we believe our benchmark provides a significant improvement over existing offline RL benchmark tasks, many of which are either saturated due to recent algorithm developments or do not cover as many of the problem dimensions as D5RL, our benchmark does have several limitations. First, we focus entirely on simulated robotics tasks. Such tasks are appealing because they cover complex dynamics and visual perception, but many aspects that make RL difficult in other domains, such as a high degree of stochasticity (e.g., in algorithmic trading) are absent in these domains. Benchmark tasks that address such domains would be very valuable and complementary to ours. Second, while our tasks reflect specific real-world systems, there is a limit to how realistic such simulated domains can be. Of course real data would be a "gold standard" in realism, but evaluating policies trained on real data would require either bridging the domain gap to simulation, or else using real physical systems, both of which would require considerable engineering and slow down the iteration cycle for algorithm developers. We therefore opted for a more conventional simulated evaluation to facilitate fast algorithms development, but we also believe that a real-world counterpart to D5RL would be valuable for the community. In conclusion, we hope that D5RL will serve as a new benchmark task for development of offline RL and online finetuning methods, and that future work can address some of the remaining blind spots of this benchmark to provide even comprehensive evaluations and facilitate more broadly applicable algorithms.

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

## A    ENVIRONMENTS

### A.1    LEGGED LOCOMOTION

We construct the locomotion tasks using MuJoCo Todorov et al. (2012) and DeepMind's dm_control Tunyasuvunakool et al. (2020) suite, using the model of Unitree's A1 quadruped from MuJoCo Menagerie Contributors (2022). The robot only receives as input proprioceptive and goal information. In particular, the robot's observations consist of its root's local forward linear velocity, orientation (roll and pitch), angular velocity (roll, pitch, and yaw), and its (12) joint angles and velocities. We also append the previous action applied. For the hiking task, we include the displacement vector between the robot to the next waypoint along the hiking path. The reward function is a simple locomotion reward that encourages a particular velocity to be tracked, subject to penalties on the body's angular velocity. For exact details on the reward function, we refer to Section IV.B of Smith et al. (2022). The robot's actions are PD targets for the 12 joints.

### A.2    STANDARD FRANKA KITCHEN MANIPULATION ENVIRONMENT

For the Standard Franka Kitchen Manipulation environment, we make some slight modifications to the Franka Kitchen environment from Gupta et al. (2019a) (RPL). The RPL Franka Kitchen environment requires controlling a simulated 9-DOF Franka Emika Robot to manipulate a set of four pre-defined objects into a desired configuration. At each timestep, a reward of 1.0 is given for each object that is in the correct configuration, with the maximum reward possible at each timestep being 4.0. The action space is joint-space control commands to the robot.

We modify the original camera angle of the RPL environment to be the camera angle used in the LEXA benchmark (Mendonca et al., 2021). Additionally, we add a wrist camera. We render both cameras at 128x128 resolutions. The observation space consists of two RGB images from the two cameras concatenated together, plus robot proprioception.

We also utilize frame stacking in our experiments. This amounts to stacking the previous three images along the channel dimension, allowing the agent to have a short history of observations from which it can estimate movement and velocity, as done in (Mnih et al., 2013).

### A.3    RANDOMIZED FRANKA KITCHEN MANIPULATION ENVIRONMENT

The Randomized Franka Kitchen Environment modifies the "Kitchenshift" domain Xing et al. (2021), which is itself a heavily modified version of the RPL Kitchen environment. The Randomized Kitchen environment includes a large degree of domain randomization and visual diversity. At the start of each episode, the initial positions of the objects are randomized, as well as textures and lighting conditions. The specific types of objects are randomized too (eg: one type of kettle can be switched for a differently shaped type of kettle). The underlying tasks and rewards are the same as in the Standard Kitchen Environment. The action space is the same as in the Standard Kitchen environment.

We use three RGB cameras (two side cameras and one wrist camera), each rendered at a resolution of 128x128 pixels. Robot proprioception is also included in the observation space. Similar to the Standard Kitchen environment, we use a frame-stacking wrapper around the Randomized Kitchen environment to maintain a history of 3 images.

### A.4    MULTI-STAGE MANIPULATION WITH SCRIPTED DATA

The multi-stage bin sorting task is an environment constructed using the DeepMind's dm_control, a software stack utilized for physics-based Simulation and RL environments. The WidowX 250 was specified with an XML file which includes information about the robot's joints with respect to their sizes and weight. A position-based controller was used for the robot, where a specified action was indicated as a change in robot position. This controller was a PID-based controller. The objects and containers were sourced from Google's Scanned Object Dataset (Downs et al., 2022), which contains photo-realistic 3D object models. From here, we selected 2 identical bins as containers and a set of objects that lie in two categories: toys and shoes. The objects were scaled to be graspable by the robot and fit in the container and are placed in the scene in any orientation (random quaternion). The

background was a static tabletop where the robot, containers, and objects were all placed as seen in Figure 4.

# B   DATASETS

We summarize the datasets, their construction and composition, for each of the tasks (organized by environment).

## B.1   LEGGED LOCOMOTION

We trained 3 A1s with the goal of tracking 3 speeds: 0.5, 0.8, and 1.0 m/s using RL (with the same inputs and reward function as described in Appendix A). We then consolidated their replay buffers and relabeled them as if their goal was to track speeds of 0.75m/s and 1.25m/s for the Interpolate Speed and Extrapolate Speed tasks, respectively. For the Hiking task, we trained a direction-conditioned policy using RL, again with the same observation and action space.

## B.2   STANDARD FRANKA KITCHEN MANIPULATION ENVIRONMENT

For our experiments we re-render the original RPL datasets (Gupta et al., 2019a) with the two cameras described in section A.2. We add in proprioception to the observations, which consists of the 9 joint angles of the robot arm. The dataset contains 563 trajectories, with $128, 569$ total transitions. The average undiscounted episode return is $261.12$, and the average number of objects manipulated per episode is $3.98$.

## B.3   RANDOMIZED FRANKA KITCHEN MANIPULATION ENVIRONMENT

We collected three distinct datasets for the Randomized Kitchen environment using tele-operation: Demonstrations, Play, and Sub-Optimal Expert. The differences between these datasets are described in Section 5.2.2. The Demonstrations dataset contains 500 total trajectories, with $250, 500$ total transitions. The average undiscounted episode return is $1148.78$, and the average number of objects manipulated per episode is $4.0$. The Play dataset contains $1,000$ total trajectories, with $501, 000$ total transitions. The average undiscounted episode return is $870.50$, and the average number of objects manipulated per episode is $3.62$. The Sub-Optimal Expert dataset contains 500 total trajectories, with $250, 500$ total transitions. The average undiscounted episode return is $911.70$, and the average number of objects manipulated per episode is $3.55$.

## B.4   MULTI-STAGE MANIPULATION WITH SCRIPTED DATA

For the Multi-Stage Bin Sorting Task, we used hand-engineered scripted policies. These scripted policies used information such as the position of the object as well as containers to solve their respective tasks of interest. These scripted policies were given a time horizon of 500 to solve this task. For the **pick and place** task, the scripted policies were constructed to randomly select one of the two objects in the scene to grasp. 70% of the time, the policy moved the object toward the correct bin. Other times, the object was directed to the incorrect bin (mimicking a scenario that the object was misclassified and sorted into the incorrect bin). For the **sorting** task, the scripted policy completed both stages of the task by picking and placing each object in succession. Our scripted policies are inspired by the procedure used in COG (Singh et al., 2020).

## C  BASELINES

### C.1  ARCHITECTURE DESIGN CHOICE

For tasks that require learning from visual observations, we utilize the Impala architecture for our experiments. For the actor and critic, the network backbone we used the architecture is found in Impala (Espeholt et al., 2018). For environments that relied on multiple camera viewpoints such as the Franka Kitchen environments, image observations were frame-stacked and passed into the network. The output of the neural network was flattened and passed through an MLP to construct the actor and critic networks for each method. For environments with a proprioceptive state, this observation was concatenated to the flattened output of the network, prior to being passed through the MLP network.

### C.2  METHODS + IMPLEMENTATION DETAILS

Here we describe the prior methods we evaluate and describe task-specific implementation details.

**IQL (Kostrikov et al., 2021)** For the implementation of IQL, we modify the open source implementation of IQL found in `https://github.com/ikostrikov/jaxrl2`. The hyperparameters utilized for both methods can be found in Table 3. During training, we utilize the data augmentations of color jitter and random crops as proposed in DrQ (Kostrikov et al., 2020) which allows for better generalization.

**CQL (Kumar et al., 2020)** and **CalQL (Nakamoto et al., 2023)** For the implementation, we modify the open source implementation of CQL found in `https://github.com/ikostrikov/jaxrl2` for CQL and CalQL. The hyperparameters utilized for both methods can be found in Table 4. During training, we utilize the data augmentations of color jitter and random crops as proposed in DrQ (Kostrikov et al., 2020) which allows for better generalization. For CalQL, the lower bound was computed with the Monte-Carlo returns calculated using the rewards of the collected demonstrations, following the recipe in Nakamoto et al. (2023).

**TD3 + BC (Fujimoto et al., 2019)** TD3+BC is an offline RL method that modifies an online RL method for the offline regime by simply adding a BC term to encourage the policy to resemble the behavior policy. For pixel-based experiments, we use the open-source implementation found in `https://github.com/ikostrikov/jaxrl2`. For state-based experiments, we use the authors' implementation at: `https://github.com/sfujim/TD3_BC`.

**RLPD (Ball et al., 2023)** RLPD is a method for online RL with access to offline data that has demonstrated state-of-the-art results on tasks designed to evaluate fine-tuning from offline RL pre-training; therefore, we include it as a main baseline for the fine-tuning regime. For 'fine-tuning' evaluation, we evaluate RLPD as designed, i.e., without pre-training. For offline evaluation, we adapt RLPD to only sample from the offline data. We use the implementation by the authors open-sourced at: `https://github.com/ikostrikov/rlpd` and use the default hyperparameters as prescribed in the paper for all environments. For the Standard Kitchen and Randomized Kitchen environments, we used an Impala network architecture for the policy and critic network encoders.

**DDPM + BC** For the implementation of DDPM+BC, we modify the implementation of the behavior cloning policy from IDQL (Hansen-Estruch et al., 2023) from `https://github.com/philippe-eecs/IDQL`. This involves attaching the convolutional encoder used to the architec-

| Hyperparameters | IQL |
|---|---|
| $\tau$ | 0.5, 0.7, 0.9, 0.95 |
| actor architecture | Impala |
| critic (Q/V) architecture | Impala |
| actor learning rate | 1e-4 |
| critic (Q/V) learning rate | 3e-4 |
| batch size | 64 |

Table 3: **Hyperparameters for IQL**. We primarily utilize the default hyperparameters as prescribed in the paper and sweep over the expectile $\tau$.

| Hyperparameters | CQL and CalQL |
|---|---|
| $\alpha$ (online + offline) | 0.1, 1, 5, 10 |
| actor architecture | Impala |
| critic architecture | Impala |
| actor learning rate | 1e-4 |
| critic learning rate | 3e-4 |
| batch size | 64 |

Table 4: **Hyperparameters for CQL and CalQL**. We primarily utilize the default hyperparameters as prescribed in the paper and sweep over the constant $\alpha$.

ture proposed in IDQL (layernorm + resnet). During training, we use data augmentations as proposed in DrQ (Kostrikov et al., 2020) which improve generalization.

Furthermore, instead of using the variance preserving schedule as used in IDQL, we use the cosine schedule (Nichol and Dhariwal, 2021) and $T = 20$. All other hyperparameters for the diffusion process and trunk architecture are the same as in IDQL. While we train for 2 gradient million steps, we recommend training for longer as the diffusion objective takes longer to train.

**IDQL (Hansen-Estruch et al., 2023)** The IDQL implementation combines the IQL implementation with the DDPM+BC implementation. After training the $Q$-function using Pixel IQL and the diffusion behavior policy using DDPM+BC, we combine the two during inference and sample the diffusion policy $N$ times and select the action that receives the highest $Q$-value. We use $N = 64$ as from IDQL, but we recommend tuning this hyperparameter as to avoid potential OOD samples.

