# OpenReview forum: "D5RL: Diverse Datasets for Data-Driven Deep Reinforcement Learning"
_ICLR.cc/2024/Conference — Submitted to ICLR 2024_

### Official Review · Reviewer_55ef · 2023-10-29

**Soundness:** 2 fair
**Presentation:** 2 fair
**Contribution:** 1 poor
**Rating:** 3
**Confidence:** 4

**Summary:**

This paper proposes a new benchmark for offline RL, which includes additional domains (such as legged locomotion and robotic manipulation) and new modalities (such as visual observation).

Several SOTA methods are also evaluated in this new benchmark, under both the pure offline setting and the online finetuning setting. Low performance is observed across domains and tasks, showing the potential space for improvements for current RL algorithms.

**Strengths:**

- This work improves D4RL by introducing some new domains and new modalities. These efforts could be helpful toward deployable offline RL, mainly for robotics.
- The benchmark provides the evaluation results for online fintuning, which has its meaning. Most of works in offline RL only utilize offline data, while neglecting the benefit of a small portion of online data to address the distribution shift issue.

**Weaknesses:**

I appreciate the works from authors to push forward the benchmarking in Offline RL. However, there still exist some critical issues for this work, that make me tend to give a strong reject:
- **The proposed benchmark does not include much more diverse domains compared to D4RL.** I think with some past works such as RoboMimic [1] and some recent works such as RL-ViGen [2], similar and even more diverse benchmarks could be easily made, by running RL agents to collect offline datasets.
- **Low technical contributions.** This is fine, but I think for papers qualified for ICLR, low-technical papers should possibly provide some new insights.
- **Possibly unfeasible settings.** As shown by evaluation results, most SOTA algorithms are not working well on this benchmark. This could possibly be attributed to the difficulties of the task, but from my perspective, there could also be some key factors that are not right, such as the problem formulation and the framework of offline RL. (This point might be not significant, but it is my personal concern about the benchmark.)
- **Overclaim of the paper title**. This work names the newly proposed benchmark as D5RL, which is obviously following the name of D4RL. However, this could be misleading for the entire community to keep pursuing the limited improvements on a new but essentially similar benchmark.

Overall, I appreciate the efforts made by the authors, and the motivation to improve the benchmarking in offline RL. This direction could be right and cherished by the community, but this work is not qualified for acceptance.


[1] Mandlekar, Ajay, et al. "What matters in learning from offline human demonstrations for robot manipulation." arXiv preprint arXiv:2108.03298 (2021).


[2] Yuan, Zhecheng, et al. "RL-ViGen: A Reinforcement Learning Benchmark for Visual Generalization." arXiv preprint arXiv:2307.10224 (2023).

**Questions:**

See `weakness`.

---

> ### Author Response · Authors · 2023-11-22
> **Thank you for your review and references!**
>
> **The proposed benchmark does not include much more diverse domains compared to D4RL.
> I think with some past works such as RoboMimic [1] and some recent works such as RL-ViGen [2], similar and even more diverse benchmarks could be easily made, by running RL agents to collect offline datasets.**
>
> We believe our tasks are significantly more diverse than RoboMimic. In particular:
> 1. Our environments use a number of different objects for the manipulation setting
> 2. We use two different types of robot arms - a Franka Emika and a WidowX, as well as an A1 quadruped robot.
> 3. The tasks we consider are more complex and involve achieving multiple objectives (i.e. arranging multiple objects in a scene).
> 4. Furthermore, our tasks have a number of continuous distribution shifts, such as scene, texture and camera randomization.
> 5. Moreover we have collected over 20 hours of realistic human manipulation data beyond pure expert demonstrations, such as play, failures, and sub-optimal execution, which was not analyzed by RoboMimic.
>
> Thank you for bringing RL-ViGen to our attention. We view this work as complementary to ours, since our manipulation environments do not overlap.
>
> 1. We aimed to create a lightweight and accessible benchmark for data-centric RL, which does not involve heavy simulators, such as CARLA and Habitat.
> 2. Our proposed environments have naturally occurring distribution shifts, such as diverse objects, visuals, lighting and camera angles, while the ones in RL-ViGen are rather artificial.
> 3. We also aimed to utilize realistic human-collected datasets, rather than replay-buffer based ones.
>
> **Possibly unfeasible settings. As shown by evaluation results, most SOTA algorithms are not working well on this benchmark. This could possibly be attributed to the difficulties of the task, but from my perspective, there could also be some key factors that are not right, such as the problem formulation and the framework of offline RL. (This point might be not significant, but it is my personal concern about the benchmark.)**
>
> We have included improved results on some of the tasks, achieved by a more extensive hyper-parameter search (see table below). The fact that SoTA RL methods achieve good performance on other benchmarks (such as D4RL) but do not achieve good performance on our benchmark is echoed by results on many real world robot manipulation tasks, in which behavior-cloning based methods tend to out-perform current RL methods. We hope that our benchmark can drive the development of RL methods which are more successful in real robot applications. For example, a recent model-based RL method [1] is able to fully solve the Standard Franka Kitchen from images with online fine-tuning.
>
> | Task Name         | BC    | DDPM + BC | IQL   | IDQL  | CQL | TD3 + BC |
> |-------------------|-------|-----------|-------|-------|-----|----------|
> | Sorting           | 32.54 | 28.47     | 33.56 | 35.59 | 0   | 14.24    |
> | Sorting Bin Noise | 44.75 | 34.58     | 45.76 | 49.49 | 0   | 21.36    |
>
>
> [1] MOTO: Offline Pre-training to Online Fine-tuning for Model-based Robot Learning
> Rafael Rafailov, et. al, Conference on Robot Learning, 2023
>
> **Overclaim of the paper title. This work names the newly proposed benchmark as D5RL, which is obviously following the name of D4RL. However, this could be misleading for the entire community to keep pursuing the limited improvements on a new but essentially similar benchmark.**
>
> We believe there are substantial differences between d4rl and our proposal. The main ones being:
> 1. Focusing on realistic robotic tasks versus simple simulated embodiments.
> 2. Learning from realistic observation spaces rather than ground-truth states.
> 3. We introduced a significant amount of diversity in terms of objects to manipulate (which require different grasps) and visual environments.
> 4. We introduced a number of new realistic human-collected datasets, such as undirected manipulation and sub-optimal execution data, rather than using RL replay buffers.

---

### Official Review · Reviewer_VAo4 · 2023-11-01

**Soundness:** 3 good
**Presentation:** 2 fair
**Contribution:** 2 fair
**Rating:** 6
**Confidence:** 3

**Summary:**

The paper introduces a new dataset benchmark for offline and online fine-tuning RL methods. The paper comprised of tasks which are of great interest to real-world robotics such as locomotion and manipulation. Also provides the diversity in domain of representation, as it covers both state-space and image domains.

**Strengths:**

* The tasks in the dataset capture various challenges that are encountered in real-world robotics such as extrapolating policies, changes in environment, different viewpoints, etc.
* Both image-based and proprioceptive modes of states are covered across settings and tasks.
* The datasets are gathered for platforms that are embodied in the real-world and hence will allow for sim-to-real transfer to some extent.

**Weaknesses:**

* Limited number of tasks for the A1 legged locomotion. Tasks such as jumping, obstacle avoidance, etc. can also be of interest in this.
* No evaluation of any of these tasks/settings in the real world. To what extent can the models trained on these datasets transfer to the real-world?

**Questions:**

Don't have any major questions

---

> ### Author Response · Authors · 2023-11-22
> **Thank you for your review**
>
> **Limited number of tasks for the A1 legged locomotion. Tasks such as jumping, obstacle avoidance, etc. can also be of interest in this.**
> Here we present an initial version of our benchmark. If there is sufficient interest in the community, it is possible to add additional locomotion tasks, including more complex embodiments in the future.
>
> **No evaluation of any of these tasks/settings in the real world. To what extent can the models trained on these datasets transfer to the real-world?**
> While we design our benchmark to reflect challenges of real world robot learning, policies trained in simulation could face challenges in real world deployment due to environment mismatch and the sim-to-real gap. The primary goal of our benchmark is not to test sim to real transfer, but rather to provide a fast and easy to use test-bed for the new offline RL algorithms, which can speed up the development and iteration cycle.

---

### Official Review · Reviewer_mGUu · 2023-11-01

**Soundness:** 2 fair
**Presentation:** 3 good
**Contribution:** 2 fair
**Rating:** 5
**Confidence:** 3

**Summary:**

The paper proposes D5RL, a novel benchmark that aims to facilitate the development of offline RL algorithms. The benchmark focuses on robotics environments based on real-world robotic systems. To address the drawbacks of existing benchmarks, D5RL includes a suite of tasks that pose different challenges for offline RL algorithms, including learning from various (sub-optimal) data distributions, image observations, temporal compositionally, generalization to randomized scenarios, and offline-to-online settings. Experiments show that existing methods that perform well on standard benchmarks fail to achieve meaningful performances on D5RL.

**Strengths:**

Overall, I think the paper is a good attempt to replace existing benchmarks that are likely saturated, and having a new benchmark would have a high impact on the offline RL community. The paper is well structured, where it first motivates the need for a new benchmark, presents a list of criterion for it, and proceeds to describe the benchmark and how it satisfies the criterion. The datasets and tasks are also well explained.

**Weaknesses:**

My biggest concern about the paper is the benchmark results section.
- My impression from Tables 1 and 2 is the proposed benchmark is too difficult for existing offline RL methods to achieve meaningful performances, since most methods except BC and IQL have near-zero returns. While having a challenging benchmark is good, a too-challenging benchmark will not provide much signal to improve the performance of current methods.
- The paper is currently lacking ablation studies to understand what components of the benchmark are challenging and/or why current algorithms fail.
- I would like to see a random policy as a simple baseline. I wonder if the benchmarked methods even do better than a random policy.
- Why is there such a big gap between IQL and other similar methods such as CQL and TD3+BC? These methods often perform very similarly in D4RL, at least in the locomotion tasks.
- The paper did not choose a good collection of RL algorithms to benchmark. There are too many value-based methods, while return-conditioned methods (or generative methods) such as DT [1] and DD [2] are not mentioned. These methods have been shown to surpass value-based methods in sparse-reward and long-horizon tasks. Online DT [3] is also a good method for the offline-to-online setting.

Other comments:
- Section 5.2 states that the goal of the Frank Kitchen environment is to study offline RL and online fine-tuning, but the following text does not mention how the proposed task facilitates online finetuning.

[1] Chen, Lili, et al. "Decision transformer: Reinforcement learning via sequence modeling." Advances in neural information processing systems 34 (2021): 15084-15097.

[2] Ajay, Anurag, et al. "Is Conditional Generative Modeling all you need for Decision Making?." The Eleventh International Conference on Learning Representations. 2022.

[3] Zheng, Qinqing, Amy Zhang, and Aditya Grover. "Online decision transformer." international conference on machine learning. PMLR, 2022.

**Questions:**

- Is there only one dataset for the legged locomotion domain? Can we have multiple datasets with various qualities similar to D4RL?
- How accurately it tracks the target speed of 0.75 m/s --> How is accuracy measured here? Is it the l2 distance between the actual and target speed?
- Is hiking a sparse-reward task?
-

---

> ### Author Response · Authors · 2023-11-22
> **Thank you for your detailed feedback!**
>
> **My impression from Tables 1 and 2 is the proposed benchmark is too difficult for existing offline RL methods to achieve meaningful performances, since most methods except BC and IQL have near-zero returns. While having a challenging benchmark is good, a too-challenging benchmark will not provide much signal to improve the performance of current methods.**
>
> Thank you for your suggestion. We have added some updated results to improve the performance on the WidowX task (see Table below), one of these challenging tasks that are referred to.
> To increase task performance, we refactored the environment by modifying the layout of the environment, the tractability of grasping objects in the scene and collecting longer demos:
>
> - **Augmenting the layout of the environment** so that there is more room to grasp the objects found in the center of the scene
>
> - **Reducing the range of the rotation of objects and the size of objects in the scene** to increase grasp success (as the simulated robot doesn’t have a rotation in gripper space)
>
> - **Collecting for a longer horizon** to ensure that the full sorting procedure occurs for the scripted sorting policy for each episode collected.
>
> | Task Name         | BC    | DDPM + BC | IQL   | IDQL  | CQL | TD3 + BC |
> |-------------------|-------|-----------|-------|-------|-----|----------|
> | Sorting           | 32.54 | 28.47     | 33.56 | 35.59 | 0   | 14.24    |
> | Sorting Bin Noise | 44.75 | 34.58     | 45.76 | 49.49 | 0   | 21.36    |
>
> We have refactored the tasks to be (1) sorting, where the agent has to sort two objects into their respective bins and (2) sorting bin noise, where the agent still has to sort the two objects but there is a probability that each object is placed into the wrong bin.
> Along with this modified task, we have included the performance of a random policy on our tasks, which achieves a 0% success rate across all tasks.
>
> **The paper is currently lacking ablation studies to understand what components of the benchmark are challenging and/or why current algorithms fail.**
>
> While we do not have ablation studies of the current RL methods, since our goal is not to develop better RL methods but rather to provide a benchmark, the benchmark tasks themselves are already designed to provide different difficulty tiers, and therefore can help researchers understand where their methods work better or worse. This was already discussed to some degree in Section 4, but we have since expanded that discussion.
> Specifically, the Standard and Randomized Kitchen environments evaluate the ability of methods to learn to compose multiple manipulation tasks over a long time horizon. The Randomized Kitchen environment additionally tests the visual robustness of methods and their ability to generalize to variations in object shape and location. The play dataset of the Randomized kitchen tests the ability of methods to learn from sub-optimal (non-demonstration quality) data.
>
> **I would like to see a random policy as a simple baseline. I wonder if the benchmarked methods even do better than a random policy.**
>
>
> Random policies achieve zero success on our manipulation environments. We have added the result to the experiment table.
>
>
> **Why is there such a big gap between IQL and other similar methods such as CQL and TD3+BC? These methods often perform very similarly in D4RL, at least in the locomotion tasks.**
>
> The fact that value-based RL methods perform worse than behavior cloning-based methods on our benchmark is an important insight/contribution, not a disadvantage. In many real robot manipulation tasks, behavior cloning-based methods tend to outperform offline value-based-RL methods, too. While value-based RL methods have achieved SoTA performance on simpler benchmarks, such as D4RL, we hope that our benchmark can help drive the development of offline RL methods that perform well on more realistic robot manipulation tasks
>
> **Section 5.2 states that the goal of the Frank Kitchen environment is to study offline RL and online fine-tuning, but the following text does not mention how the proposed task facilitates online finetuning.**
>
> As far as we are aware no current method can solve the environment fully with only the provided offline data. Because of this, additional online fine-tuning is a necessary step to be able to learn to fully solve it. We hope that our benchmark can provide a good way to evaluate methods for offline-to-online fine-tuning, where the available offline data is insufficient to fully solve a task, but can provide a good starting off point for initializing a policy for online learning. We carry online fine-tuning experiments on a number of our environments, results are shown in Table 2.

---

### Official Review · Reviewer_axPt · 2023-11-14

**Soundness:** 2 fair
**Presentation:** 2 fair
**Contribution:** 2 fair
**Rating:** 5
**Confidence:** 4

**Summary:**

The submission proposes a new benchmark for offline RL algorithms.
The benchmark modifies previous domains or adds new domains to evaluate unsolved challenges in offline RL.
Experiments demonstrate that existing offline RL algorithms do not perform well for the tasks in the benchmark.

**Strengths:**

The introduction of new domains and offline datasets in the benchmark offers the community a broader range of choices, potentially supporting the easier development of offline algorithms.

**Weaknesses:**

It is unclear whether promoting the use of this benchmark within the community will indeed accelerate offline RL algorithm development.
- the benchmark does not *specifically* address current challenges in offline RL. While the paper broadly covers various challenges of offline RL,
  - some of these challenges could be already observed in previous benchmarks. Evaluating temporal compositionality seems  possible using previous benchmarks (D4RL Maze2D, AntMaze, FrankaKitchen-Mixed, Calvin). The proposed benchmark has less diversity than D4RL, and different levels of data distribution are also discussed in D4RL. However, the text does not explicitly claim why using this new benchmark is better than using previous benchmarks for each challenge.
To encourage using this benchmark, the paper should address questions like: when to choose this benchmark? Are previous works successful on prior benchmarks likely to fail on this new benchmark specifically due to one of specific challenges?
  - addressing offline-to-online learning adaption appears to be incremental. As almost all previous offline RL benchmarks are based on simulation environment and rule-based reward function, implementing online fine-tuning on the top of exisiting benchmarks is feasible.
  - More information or resource should be provided to challenge the community to solve the image observation version of Franka Kitchen. Is the failure of existing works attributed to the incompleteness of offline RL algorithms or the lack of image representation learning? Can this task be intuitively completed solely with the provided offline data? If not, is there an alternative data source that the community can leverage to address this challenge?

- Even if the benchmark doesn't specifically address one of those challenges, it can still be beneficial to have a domain with a collection of challenges if it is important to solve the domain. However, the provided domains are not significantly more realistic or different compared to existing tasks, making them less of an important target domain that the community aims to solve.

**Questions:**

Questions are included in weakness section.

---

> ### Author Response · Authors · 2023-11-22
> **Thank you for your detailed review!**
>
> **Some of these challenges could be already observed in previous benchmarks. Evaluating temporal compositionality seems possible using previous benchmarks (D4RL Maze2D, AntMaze, FrankaKitchen-Mixed, Calvin).**
>
> We agree that these environments test temporal compositionality to some degree. However, performance on the D4RL maze tasks is becoming saturated. While the existing Franka and Calvin environments still pose a challenge, they do not test other scenarios, such as the multimodality of the data distribution (Franka kitchen is composed entirely of demonstration data, and CALVIN is composed entirely of play data). In our “play” datasets tasks can be executed partially and in arbitrary order and the agent needs to learn how to compose these together, which proves to be quite challenging.
>
> **The proposed benchmark has less diversity than D4RL, and different levels of data distribution are also discussed in D4RL.**
>
> In this proposed benchmark we focus on realistic robotics tasks. While it is true that D4RL considered different levels of data quality, it did so by using different parts of an online RL replay buffer. Instead in this work we focus on more realistic data distributions, such as sub-optimal demonstrations and play data.
>
> **The text does not explicitly claim why using this new benchmark is better than using previous benchmarks for each challenge. To encourage using this benchmark, the paper should address questions like: when to choose this benchmark? Are previous works successful on prior benchmarks likely to fail on this new benchmark specifically due to one of specific challenges?**
>
> We cover the difference between d5rl and other benchmarks in our Related Works section. In particular we believe it addresses the following challenges:
>
> 1. Uses models of real robot systems.
> 2. Uses realistic datasets, such as demonstrations with varying quality provided by haman tele-operators, play data and scripted planners.
> 3. Addresses realistic challenges, such as learning from high-dimensional observations and robustness to distribution shifts.
> 4. Compared to other works we believe it strikes a better balance addressing the above challenges and time/complexity/computational requirements.
>
>
> **Addressing offline-to-online learning adaption appears to be incremental. As almost all previous offline RL benchmarks are based on simulation environment and rule-based reward function, implementing online fine-tuning on the top of exisiting benchmarks is feasible.**
>
> We do not claim that our work proposes or formalizes the online fine-tuning problem. Instead we aim to demonstrate that our environments and datasets are suitable for that application.
>
> **More information or resource should be provided to challenge the community to solve the image observation version of Franka Kitchen. Is the failure of existing works attributed to the incompleteness of offline RL algorithms or the lack of image representation learning? Can this task be intuitively completed solely with the provided offline data? If not, is there an alternative data source that the community can leverage to address this challenge?**
>
> The kitchen environments are quite challenging due to a variety of factors. As far as we are aware at the time of submission no offline RL or online fine-tuning approach had fully solved even the fully observable low-dimensional state versions. However, the standard (non-randomized) environments are feasible to solve, since a recent model-based work provided a complete solution [1].
>
> **Even if the benchmark doesn't specifically address one of those challenges, it can still be beneficial to have a domain with a collection of challenges if it is important to solve the domain. However, the provided domains are not significantly more realistic or different compared to existing tasks, making them less of an important target domain that the community aims to solve.**
>
> We believe that the challenges posed by the proposed environments are significant and indicative of challenges of real world robot learning. In particular issues such as visual diversity and robustness remain quite challenging for real world robot systems.
>
>
> [1] MOTO: Offline Pre-training to Online Fine-tuning for Model-based Robot Learning
> Rafael Rafailov, et. al, Conference on Robot Learning, 2023

---

### Meta-Review · Area_Chair_ffrK · 2023-12-04

**Metareview:**

This paper proposes a new benchmark for offline RL that focuses on realistic simulations of robotic manipulation and locomotion environments.

**Reviewers have reported the following strengths:**

- Interest to the community.

**Reviewers have reported the following weaknesses:**

- Technical contribution;
- Lack of problems;
- Benchmarking.

**Decision**

This paper has received unanimous negative reviews. Reviewers appreciate the potential interest that this work could have on the community, but pointed out several critical issues that should be addressed, in particular about the benchmarking and diversity of the datasets.

**Justification For Why Not Higher Score:**

N/A

**Justification For Why Not Lower Score:**

N/A

---

### Decision · Program_Chairs · 2024-01-16

Reject